# Studies of Fracture Damage Caused by the Proppant Embedment Phenomenon in Shale Rock

**Mateusz Masłowski \*, Piotr Kasza, Marek Czupski, Klaudia Wilk and Rafał Moska**

Oil and Gas Institute – National Research Institute, Lubicz 25A Str., 31-503 Krakow, Poland;
piotr.kasza@inig.pl (P.K.); marek.czupski@inig.pl (M.C.); klaudia.wilk@inig.pl (K.W.); rafal.moska@inig.pl (R.M.)
\* Correspondence: mateusz.maslowski@inig.pl

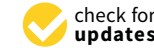

**Featured Application: The result of laboratory imaging of the embedment phenomenon may be one of the preliminary assessments of the effectiveness of hydraulic fracturing at the design stage.**

**Abstract:** This paper concerns the effect of proppant embedment related to hydraulic fracturing treatment. This phenomenon occurs if the strength of a dry reservoir rock is lower than that of proppant grains. The aim of this research was the laboratory determination of the loss of width of the proppant pack built of light ceramic grains. A laboratory simulation of the embedment phenomenon was carried out for a shale rock on a hydraulic press in a heated embedment chamber specially prepared for this purpose. Tests were conducted at high temperature and axial compressive stress conditions. The surfaces of cylindrical core plugs (fracture faces) were imaged under an optical microscope equipped with 3D software. The fracture faces were examined and compared before and after the embedment phenomenon. Analysis of the obtained images of the fracture face was done, based on a research method of the embedment phenomenon developed at the Oil and Gas Institute—National Research Institute. On the basis of the laboratory tests, the parameters characterizing the embedment phenomenon were defined and discussed. In addition, the percentage reduction in the width of the proppant pack was determined.

**Keywords:** embedment; shale rock; proppant pack; fracture width

## 1. Introduction

Hydraulic fracturing is one of the main methods for stimulating unconventional hydrocarbon reservoirs. In the case of shale formations, which are characterized by increased content of clay minerals, for intensification treatments to be effective, numerous fractures and cracks should be created [1–7], as shown in Figure 1 [2,8–10].

The producing formation is fractured using hydraulic pressure, and then proppants are pumped into the fractures with a fracturing fluid [11]. The industry has been making use of slickwater, where the proppant transport is governed by the high velocity of the injected water, unlike polymer-based fluids for which the transporting mechanism is based on viscosity [12]. The literature [12–14] has reported a significant use of hybrid technologies that combine slickwater and polymer fluids. In hydraulic fracturing, energized fluids are also used (fluids with one compressible component such as nitrogen or carbon dioxide) [15]. The use of a gas component helps to reduce the hydrostatic pressure. It also supports wellbore and fracture clean up. Polymer-based fluids are still the most commonly used type of fracturing fluids [12]. The material used for proppants can range from natural sand grains called frac sand and resin-coated sand to high-strength ceramic materials and resin-coated ceramic materials [11].

The typical proppant sizes in shale reservoirs hydraulic fracturing are generally between 30 and 50 mesh (from 0.300 mm to 0.600 m) and between 40 and 70 mesh (from 0.212 mm to 0.420 mm).

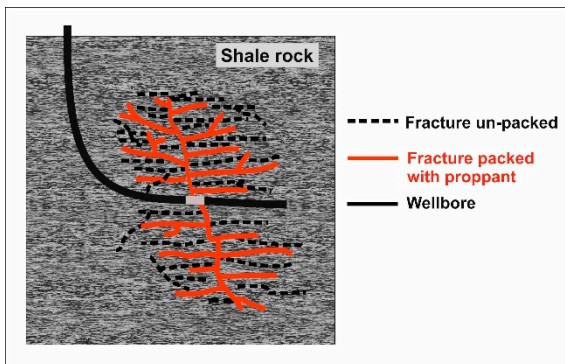

**Figure 1.** Visualization of numerous fractures and microcracks that allow an absorbed gas to be release from the shale rock.

Numerous fractures and cracks created in shale formations are characterized by low width and large values of the stimulated reservoir volume (SRV). It must also be pointed out that a proper selection of the proppant should ensure high conductivity of the whole generated system of fractures distributing the proppant to the furthest parts of the fractures [11,16,17]. Apart from the way by which the proppant is transported and placed in the fractures, the phenomena presented in Figure 2 have a significant influence on the effective packaging of fractures with the proppant [10,17]. This occurs after the treatment, when compressive stress closes the fracture on the proppant.

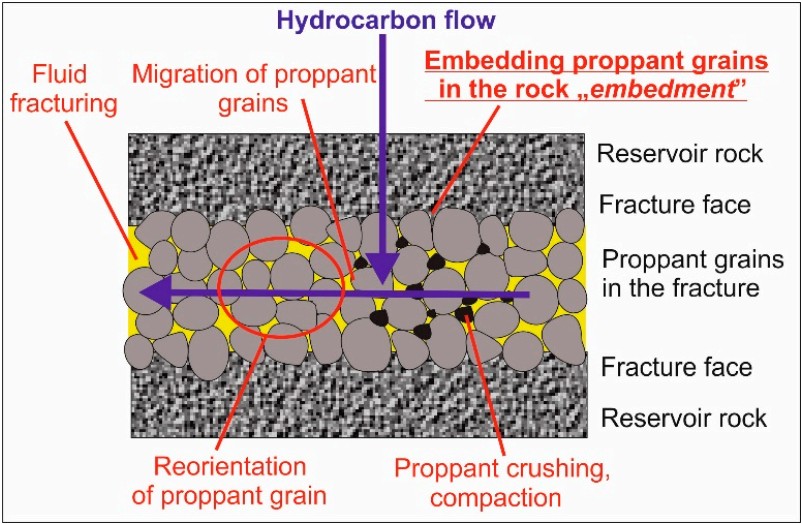

**Figure 2.** The phenomena influencing the effective packaging of a fracture and the achievement of high conductivity.

The proppant embedment phenomenon presented in this model causes the decrease of the width $W_f$ of the created fracture (Figure 3) [7,10,18–21] and an increase in the damage of the fracture face, which results in a decrease of its permeability and conductivity [22–25].

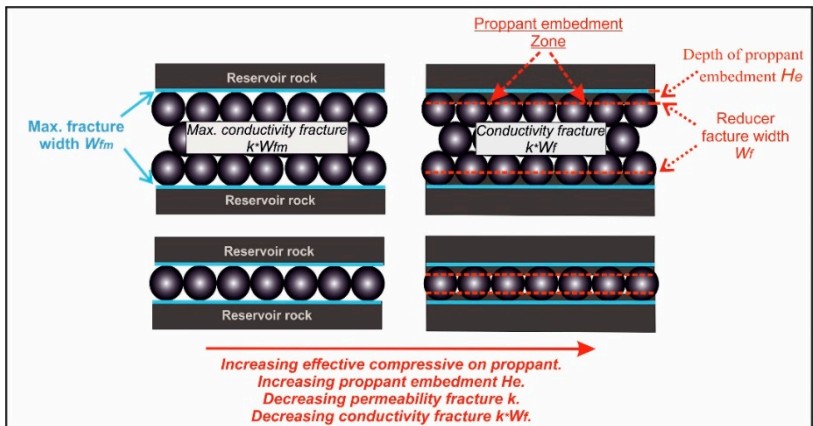

**Figure 3.** The effect of the embedment phenomenon on the width of a packed fracture for various proppant surface concentration.

For many years, a number of laboratory tests, imaging tests, and mathematical modeling of the embedment phenomenon have been performed. They are constantly being modernized as the capabilities of hardware and software increase.

In a research project [23], the fracture surface strength and fracture conductivity for American shale reservoirs (Barnett, Haynesville, Marcellus) were studied, and the kneading depth of small proppant grains into the fracture face was determined [23,26]. In the mentioned study, long-term conductivity of the dry and pre-saturated fracture was determined.

A study of proppant embedment in shales and its effect on hydraulic fracture conductivity is present also in the literature [4]. It described the relations between rock mineralogy, mechanical properties, fluid composition, and proppant embedment. Initial results showed a close correlation between the amount of proppant embedment at a given stress and the rock stiffness, which is affected by its mineral content, mainly, the amount and type of clay minerals. These correlations are used to predict the amount of conductivity loss due to proppant embedment in different unconventional reservoirs [4].

Another study [27] presents proppant embedment, which occurs in rock formations and can result in rapid decline of hydrocarbon production. These studies highlight the importance of the creep phenomenon (a function of confinement and temperature), of the percentage of clay content, and of the surface roughness in proppant embedment. Other parameters, such as time, temperature, and fracture fluid, can also impact the rate of proppant embedment. Also presented are numerical and analytical models representing proppant embedment [27].

The purpose of the laboratory tests presented in this paper was to determine the quantities characterizing the embedment phenomenon for dry polish shale rock (i.e., the total depth and width of the dents of proppant grains in the fracture face). The obtained values allowed to determine the percentage damage of the surface of the fracture face, the fracture width packed with proppant, and the percentage reduction in the fracture width, under given conditions at high temperature and axial compressive stress. The analyses of the obtained images of the fracture face were done on the basis of the research method of the embedment phenomenon developed at the Oil and Gas Institute-NRI.

## 2. Materials and Methods

### 2.1. Characteristics of the Reservoir Rock and Proppant Material Used for Testing

Shale rock (Figure 4a) containing 47.7% of clay minerals was used for testing. The content of quartz amounted to 24.4%, that of carbonates to 14.2%, and that of other components to 13.7%. A lightweight ceramic proppant 30/50 mesh (Figure 4b) with a grain size from 0.600 to 0.300 mm was used as the

proppant material. The mean diameter of the proppant grains was 0.450 mm. The roundness and the sphericity of the grains was 0.9, the bulk density was 1.51 g/cm$^3$.

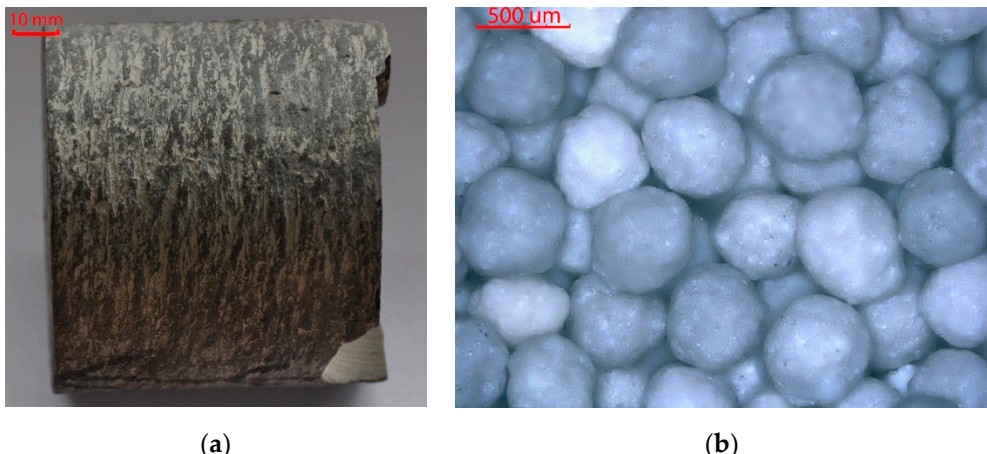

(**a**)　　　　　　　　　　　　　　　　　　　　　　(**b**)

**Figure 4.** Materials used for testing: (**a**) Shale rock; (**b**) Proppant.

### 2.2. Methodology for Studying the Embedment Phenomenon

The research methodology developed in the Oil and Gas Institute – NRI was used [7,10,20,21,28]. It allowed an initial determination of the primary roughness of the fracture face. It was determined for several selected areas, and then the average roughness was calculated from the roughness profiles along the selected measurement sections. The method of determination of the surface roughness and the measurements are presented in Figure 5 [7,10,20,21]. Equation (1) was used [7,10,20,21,28].

$$R = \frac{\sum_{i=0}^{n} H_{p_i} + \sum_{i=0}^{n} H_{v_i}}{n_p + n_v} \tag{1}$$

where $R$ is the roughness of the profile surface along the measurement section (mm), $H_p$ is the peak height (mm), $H_v$ is the valley depth (mm), $n_p$ is the total number of peaks (-), $n_v$ is the total number of valleys (-).

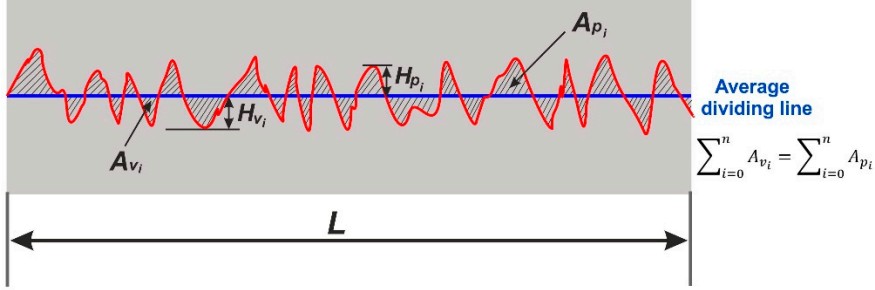

**Figure 5.** An example of the surface roughness profile along the measurement section for the selected area on the surface of the fracture face.

The average primary roughness $R_a$ for the entire surface of the fracture face was determined as an arithmetic average of the roughness of the profiles determined for the individually selected areas.

Laboratory simulation of the embedment phenomenon consisted of placing a proppant between two cylindrical core plugs and then exposing it to the set axial compression stress, at the set temperature, for the set period of time (Figure 6a,b) [7,10,21,22].

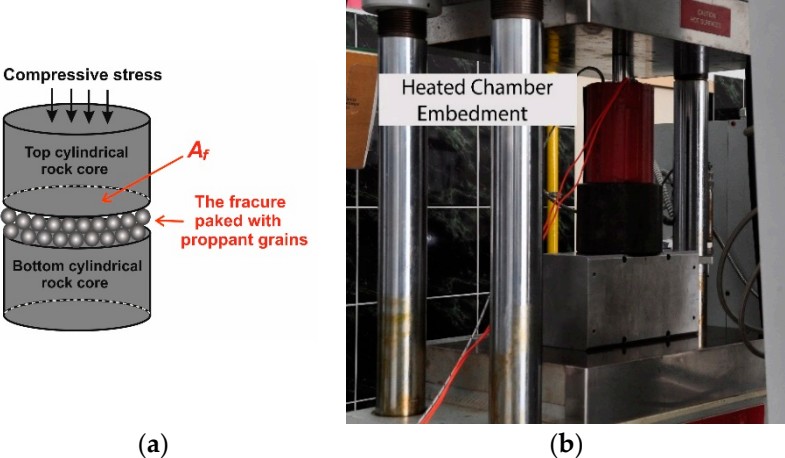

(**a**)                                       (**b**)

**Figure 6.** Test unit: (**a**) Arrangement of cores and proppant into the chamber; (**b**) Hydraulic press with heating chamber for the simulation of the embedment phenomenon.

The amount of the proppant material needed to pack the fracture and obtain the specified surface concentration was determined according to Equation (2) [10,28,29]:

$$m_p = A_f \cdot 10^{-1} \cdot C \tag{2}$$

where $m_p$ is the weight of the proppant (g), $C$ is the surface concentration of the proppant (kg/m$^2$), $A_f$ is the surface area of the fracture face subjected to compression stress (cm$^2$).

The analysis of the fracture face after simulation of the embedment phenomenon consisted in the determination of the average depth of embedment of the proppant grains and of the damage of its surface. The method of determination of embedment depth and damage of the fracture face along the measurement section is presented in Figure 7 [7,10,20,21,26] and in Equation (3) [7,10,20,21,28].

$$H_e = \frac{\sum_{i=0}^{n} H_{e_i}}{n_e} \tag{3}$$

where $He$ is the average depth of proppant embedment in the fracture face of the profile along the measurement section (mm), $He_i$ is the valley depth (embedment of a proppant grain in the fracture face) (mm), $n_e$ is the total number of valleys (embedment of proppant grains in the fracture face) (-).

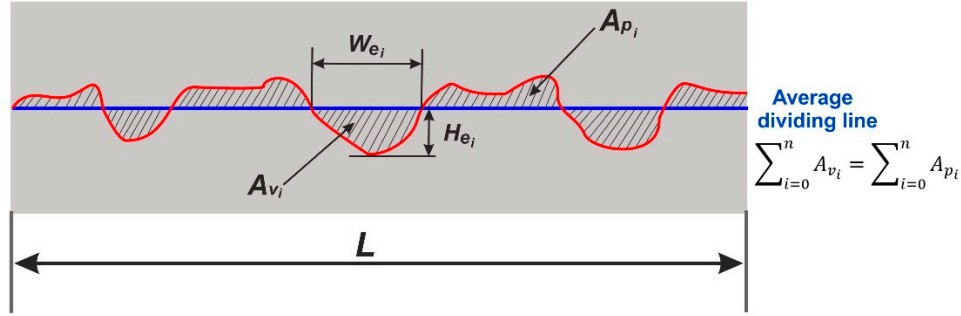

**Figure 7.** Sample profile of the depth and width of grain embedment (valleys) along the measurement section in the selected area, on the surface of the fracture face.

The total average depth $He_t$ of proppant embedment in the fracture faces (rock), expressed in mm, was determined according to Equation (4) [7,10,20,21,28]:

$$H_{e_t} = H_{e_{T.a}} + H_{e_{B.a}} \tag{4}$$

where $He._{T.a}$ is the average depth of proppant embedment in the top fracture face, corresponding to the arithmetic average of the obtained values for individually specified areas (mm), $He._{B.a}$ is the average depth of proppant embedment in the bottom fracture face, corresponding to the arithmetic average of the obtained values for individually specified areas (mm).

The percentage damage of the fracture surface ($PDW_e$) for the profile, along the measurement section was determined according to Equation (5) [7,10,20,21], expressed in (%):

$$PDW_e = \frac{\sum_{i=0}^{n} W_{e_i}}{L} \cdot 100 \tag{5}$$

where $W_{e_i}$ is the valley width, i.e., the embedment of a proppant grain in the fracture face (mm), and $L$ is the length of the measurement section (mm).

The total percentage damage of the fracture surface $PDW_{e_t}$ (embedment of the embedding proppant grains on the surface of the fracture faces) was determined according to Equation (6) [7,10,20,21], expressed in (%):

$$PDW_{e_t} = \frac{PDW_{e_{T.a}} + PDW_{e_{B.a}}}{2} \tag{6}$$

where $PDW_{e_{T.a}}$ is the average percentage damage of the surface of the top fracture face (rock), corresponding to the arithmetic average of the obtained values for individually specified areas (%), and $PDW_{e_{B.a}}$ is the average percentage damage of the surface of the bottom fracture face (rock), corresponding to the arithmetic average of the obtained values for individually specified areas (%).

The effect of the embedment phenomenon on the effective width of the fracture packed with proppant after exposure to axial compression stress was determined using Equations (7) and (8) [7,10, 20,21,28]:

$$W_f = W_{f_m} - H_{e_t} \tag{7}$$

where $W_f$ is the fracture width packed with proppant, taking into account the embedment phenomenon (mm), and $W_{fm}$ is the maximum fracture width packed with proppant, without the occurrence of the embedment phenomenon (mm).

The percentage reduction of the fracture width ($PRW_f$) packed with proppant, taking into account the embedment phenomenon, was determined according to Equation (8) [7,10,20,21], expressed in (%):

$$PRW_f = \frac{H_{e_t}}{W_{f_m}} \cdot 100 \tag{8}$$

The maximum width $W_{fm}$ of the fracture packed with proppant, without the occurrence of the embedment phenomenon, was determined according to the research procedure previously mentioned in this paper. The only difference was the use of cylindrical steel plugs instead of cylindrical core plugs, which have a steel hardness of more than 43 on the Rockwell C scale (HRC). The maximum width $W_{fm}$ of the fracture packed with proppant was measured throughout the testing with the use of an LVDT (Linear Variable Differential Transformer) device. LVDT readings took into account the amount of deformation of the test unit (i.e., hydraulic press, measuring chamber, and steel plugs) under the specified conditions of axial compressive stress and temperature.

## 3. Execution of a Laboratory Simulation of the Embedment Phenomenon and Analysis of the Obtained Test Results

The tests were performed on cylindrical core plugs with a diameter of 2.54 cm. Firstly, the average primary roughness $Ra$ of the entire surface of the core plug face (for the top and bottom fracture face), presented in Figure 7, was determined according to the test procedure described in the previous part of the paper. It was determined as an arithmetic average of two selected areas on the face of the tested core plug, from one profile running across the tested area. These tests were performed using an optical microscope (Figure 8), and the results are presented in Figures 9 and 10.

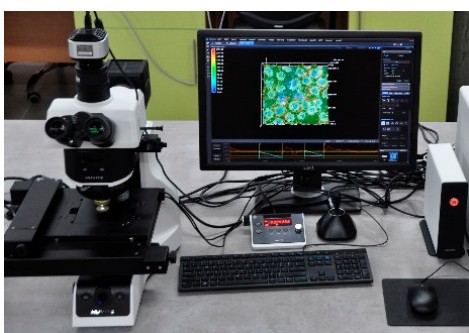

**Figure 8.** Optical microscope with 3D software.

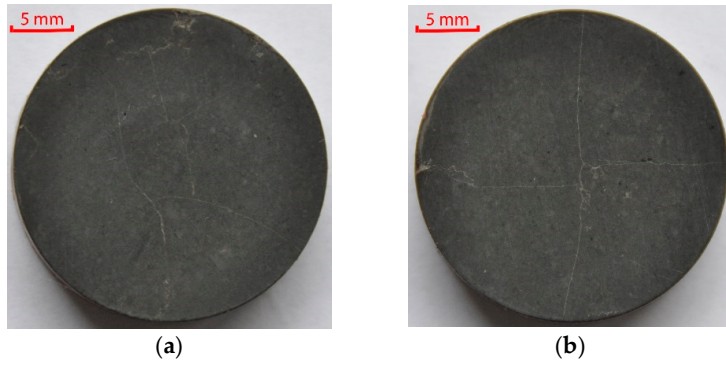

(**a**) (**b**)

**Figure 9.** Surface of the core plug sample with a diameter of 2.54 cm before the embedment test: (**a**) Top; (**b**) Bottom.

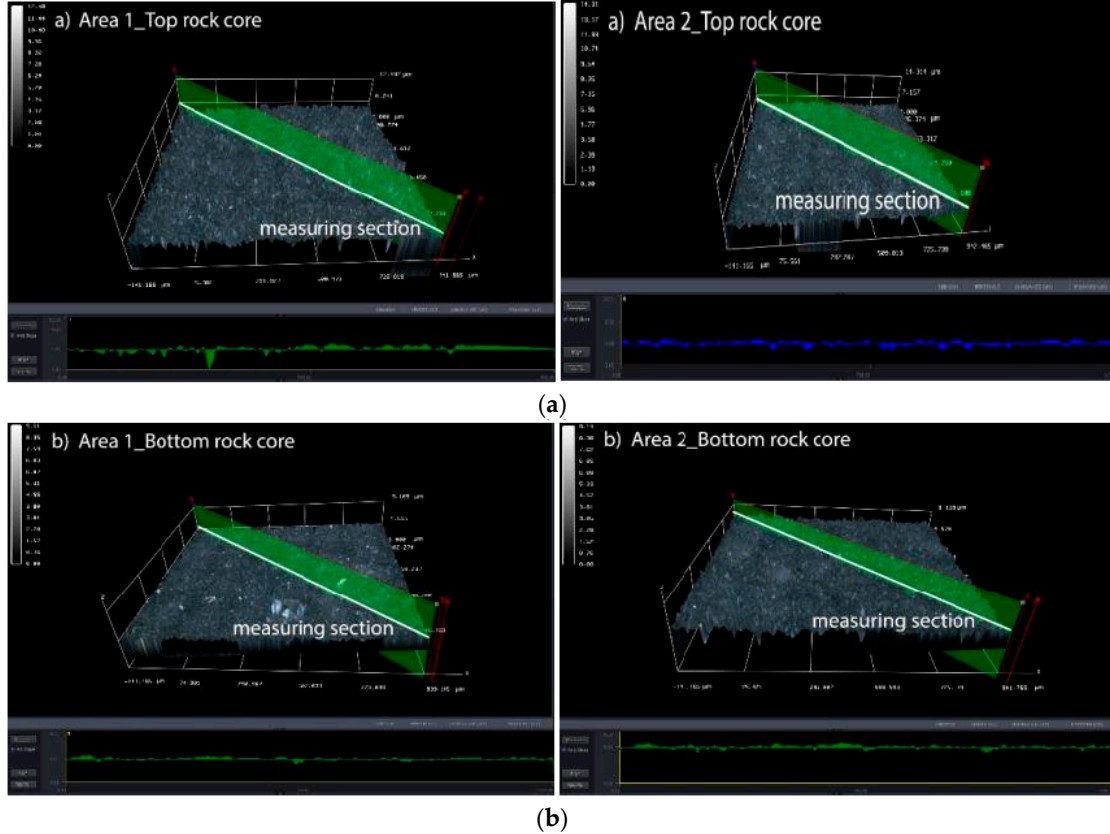

**Figure 10.** Determination of the primary roughness of the core plug face (fracture faces): (**a**) Top; (**b**) Bottom.

The average primary roughness $R_a$ of the entire face of the top core plug amounted to 0.00066 mm +/− 0.00015 mm. For the bottom core plug, it amounted to 0.00033 mm +/− 0.00007 mm.

Next, a laboratory simulation of the phenomenon of proppant embedment in the fracture faces, on the test unit presented in Figure 6, was carried out.

The test conditions are presented in Table 1.

**Table 1.** Conditions for the tests no. 1 and no. 2.

| Conditions for Test | |
|---|---|
| **Temperature, (°C)** | 70.0 |
| Surface concentration of proppant, (kg/m$^2$) | 2.44 |
| Compressive stress, (MPa) | 48.3 |
| Exposure to the defined compressive stress, (hours) | 6 |

The result of test no. 1 is presented in Figures 11–13 and in Tables 2 and 3.

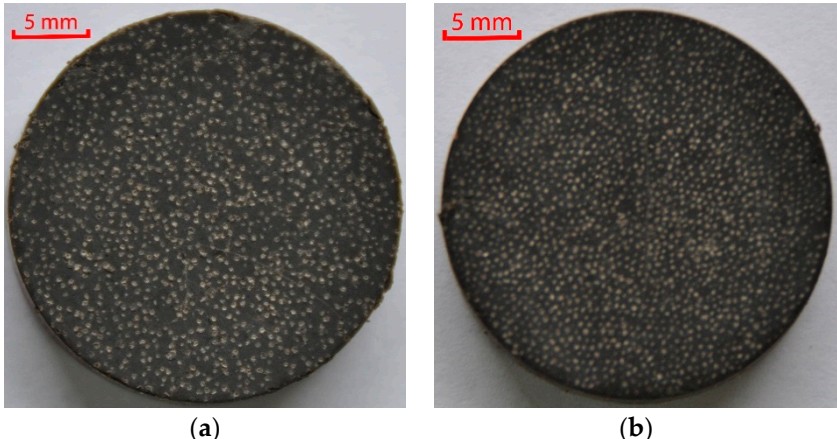

(**a**)    (**b**)

**Figure 11.** Surface of the core plug sample with a diameter of 2.54 cm after the embedment test: (**a**) Top; (**b**) Bottom.

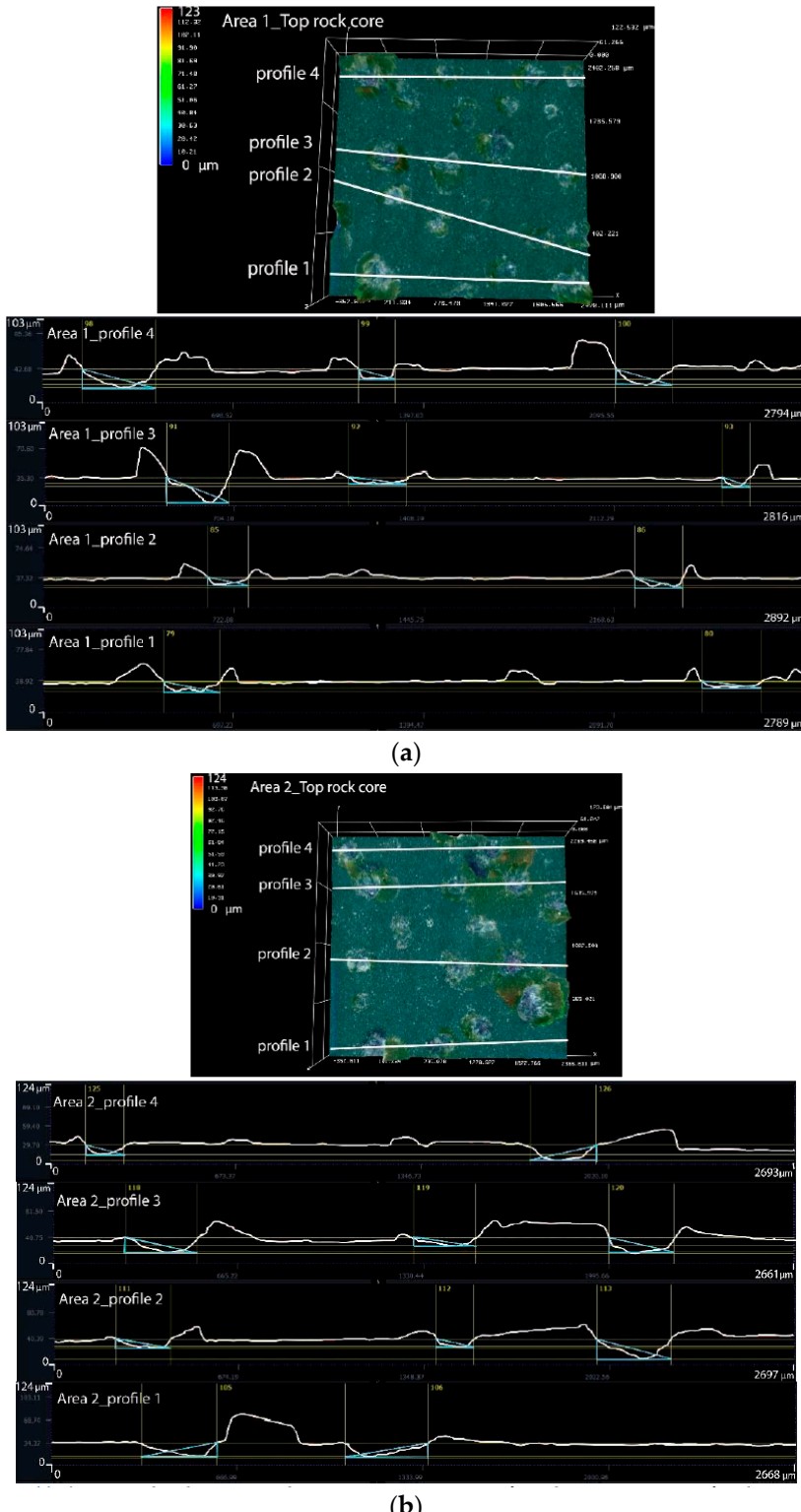

**Figure 12.** Average depth $H_{e_{T,a}}$ and average percentage surface damage $PDW_{e_{T,a}}$ for the top core plug (Test 1): (**a**) Area 1; (**b**) Area 2.

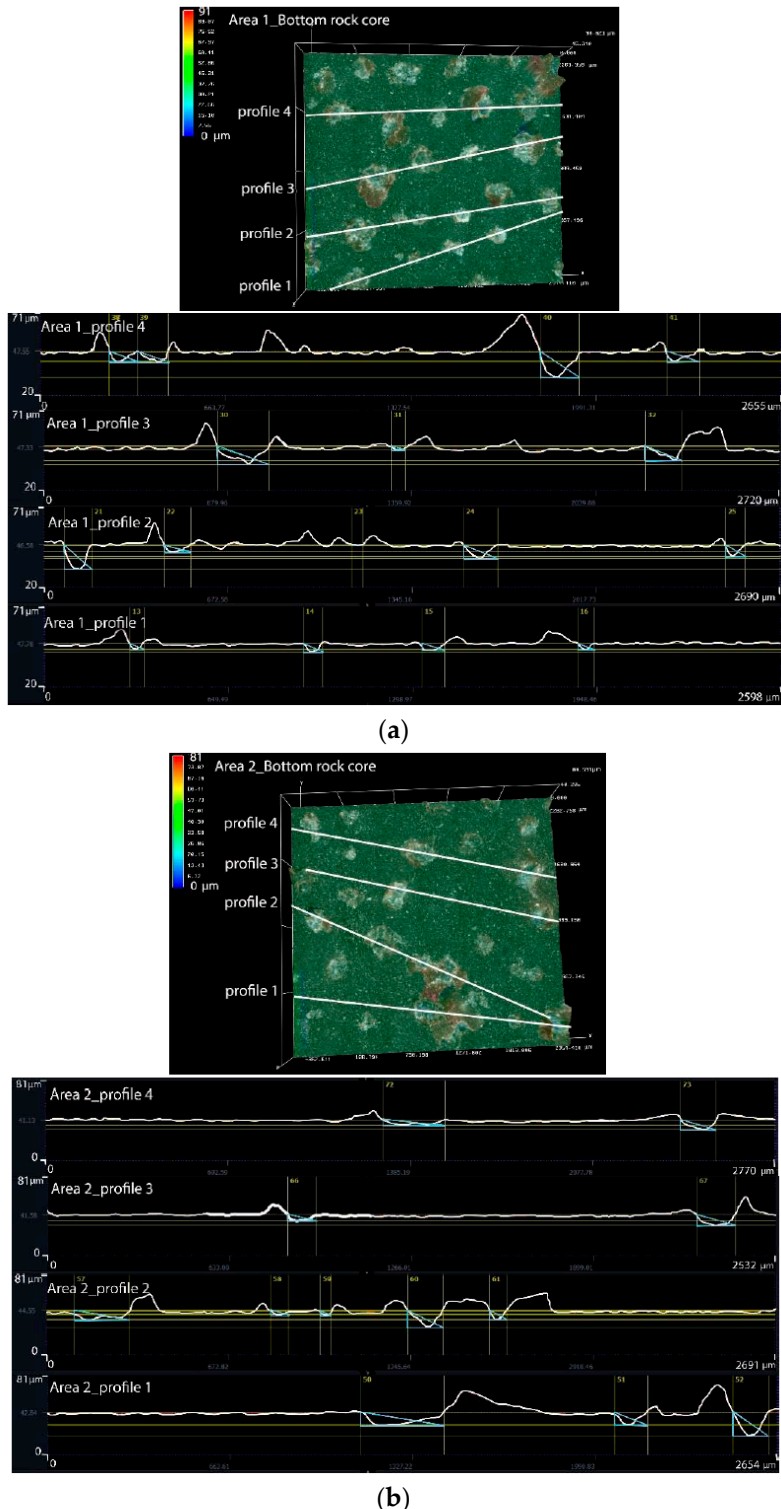

**Figure 13.** Average depth $H_{e_{B.a}}$ and average percentage surface damage $PDW_{e_{B.a}}$ for the bottom core plug (Test 1): (**a**) Area 1; (**b**) Area 2.

**Table 2.** Total average depth of proppant embedment in the fracture faces—Test no. 1.

| Fracture Face | No. of the Tested Area | Surface Area (mm²) | Average Measurement Section Length (mm) | Total Measurement Sections Length (mm) | $H_e$ (mm) | $He_a$ (mm) | $He_t$ (mm) |
|---|---|---|---|---|---|---|---|
| Top | 1 | 7.2188 | 2.8230 | 11.2919 | 0.0141 | 0.0170 | |
| | 2 | 6.8904 | 2.6717 | 10.6866 | 0.0200 | | 0.0254 |
| Bottom | 1 | 6.8251 | 2.6658 | 10.6632 | 0.0065 | 0.0084 | |
| | 2 | 6.9765 | 2.6620 | 10.6481 | 0.0102 | | |

**Table 3.** Average percentage damage of the fracture surface—Test no.1.

| Fracture Face | No. of the Tested Area | $W_e$ (mm) | $PDW_e$ (%) | $PDW_{e_a}$ (%) | $PDW_{e_t}$ (%) |
|---|---|---|---|---|---|
| Top | 1 | 1.9268 | 17.1 | 19.2 | |
| | 2 | 2.2712 | 21.2 | | 17.1 |
| Bottom | 1 | 1.5250 | 14.3 | 14.9 | |
| | 2 | 1.6647 | 15.6 | | |

Test no. 2 was performed in order to determine the maximum achievable width of the fracture packed with a light ceramic proppant without embedment. The test took into account the width reduction which can occur as a result of proppant crushing and proppant grains rearrangement within the fracture. The conditions of test no. 2 are presented in Table 1. In test no. 2, cylindrical core plugs were replaced with cylindrical steel plugs.

After 6 hours of exposure to the defined axial compressive stress, a maximum fracture width $W_{fm}$ of 1.514 mm was obtained.

The uncertainty of the estimated width of the fracture packed with proppant was determined on the basis of the accuracy of the LVDT fracture gauge +/- 0.001 mm. The uncertainty of the estimated total average depth of proppant embedding in the fracture faces was determined on the basis of the standard deviation from the average value. The parameters of the fracture effectively packed with the proppant are presented in Figure 14.

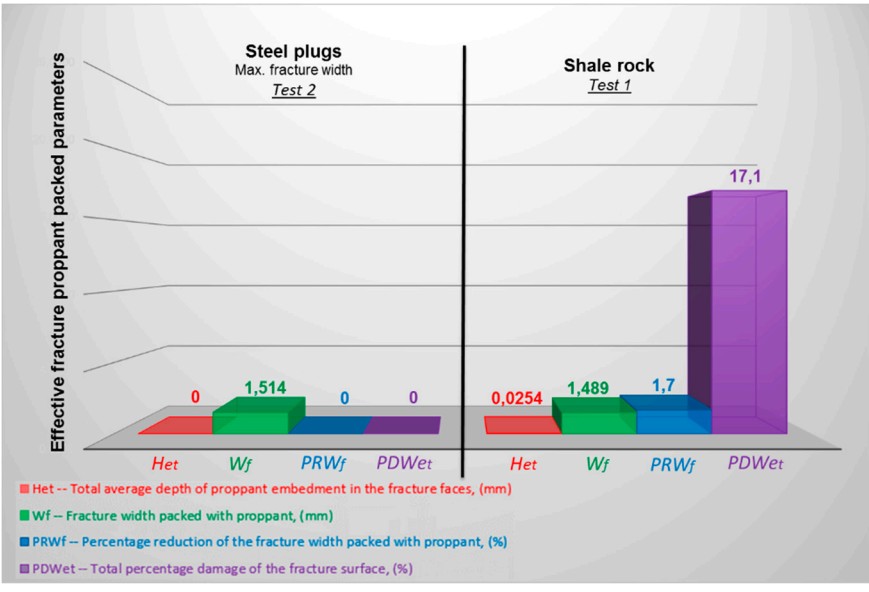

**Figure 14.** Parameters of the fracture effectively packed with the proppant for cylindrical steel plugs and shale rock.

In addition, an attempt was made to simulate the effect of embedment on an effectively packed fracture with only one proppant layer. The maximum fracture width, corresponding to the average diameter of the tested proppant grains, amounting to 0.450 mm, was used for the calculation. It was assumed that the value of the average depth of proppant embedment $He_t$ and surface damage $PDWe_t$, were equal to the values obtained in test no. 1. The results of this analysis are presented in Figure 15.

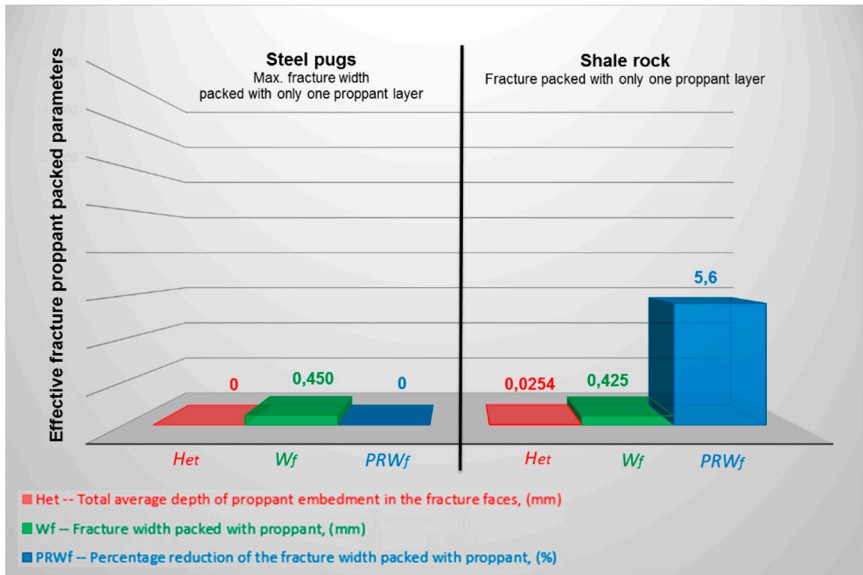

**Figure 15.** Parameters of the fracture effectively packed with the proppant for shale rock packed with one layer of proppant (grains size from 0.600 to 0.300 mm).

## 4. Discussion

Measurements were performed for a light ceramic proppant consisting of grains with size from 0.600 to 0.300 mm, with a low surface concentration of proppant of 2.44 kg/m$^2$ (several layers of proppant grains), and axial compression stress of 48.3 MPa for 6 hours at 70 °C. The average diameter of the proppant grains was 0.450 mm. The tested dry shale rock was characterized by:

1. On the basis of the analyzed embedment profiles, it was concluded that 17.1% of the total surface was damaged by the proppant grains.
2. The total depth of proppant embedment in the fracture faces was 0.0254 mm.
3. The obtained width of the fracture was 1.489 mm, therefore 1.7% less than the maximum achievable fracture width, which could be 1.514 mm, for the specified test conditions.

For the additionally simulated maximum fracture width (0.450 mm), corresponding to only one layer of the tested proppant, a decrease of the maximum fracture width by 5.6% was obtained. In this case, the depth of the proppant embedment of 0.0254 mm was used for the calculation. The final width of such packed fracture was 0.425 mm.

The size of the fracture width determines the flow of hydrocarbons through the fracture packed with proppant grains to the wellbore.

The tested dry shale rock allowed to maintain the width of the packed fracture in order for hydrocarbons to flow.

Test results indicate that the developed method of measurement may be used for preliminary assessments when choosing the proppant type and fracturing fluid for hydraulic fracturing of unconventional reservoirs, especially shale rocks.

**Author Contributions:** Conceptualization, M.M., P.K., M.C., K.W. and R.M.; Formal analysis, M.M., P.K., M.C., K.W. and R.M.; Funding acquisition, M.M.; Investigation, M.M. and P.K.; Methodology, M.M.; Software, M.M.;

Supervision, P.K., M.C. and K.W.; Validation, M.M. and R.M.; Visualization, M.M.; Writing—Original Draft, M.M.; Writing—Review & Editing, P.K., M.C. and K.W.

**Funding:** This paper is based on the results from statutory work. Archive no. DK-4100-56/17.

**Conflicts of Interest:** The authors declare no conflict of interest.

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
