# Peer review of "Studies of Fracture Damage Caused by the Proppant Embedment Phenomenon in Shale Rock"

_applsci, doi:10.3390/app9112190_

Round 1

Reviewer 1 Report

Studies of fracture damage caused by proppant embedment phenomenon in shale rock

The theme of the paper fits very well with this scope of the journal, and it is helpful to broaden the influence of this international journal. This paper provides a clear insight into this field of study by incorporating an experimental approach to understand the impact of proppant embedment on fracture width reduction. However, authors need to carry out the following improvements to the manuscript in order to achieve the publishing stage.

Comments:

1)      Fig. 1 is not clear.

2) Width reduction can occur as a result of proppant crushing and proppant rearrangement within the fracture. Did you account these factors in your calculations? Please explain as it is not clear.

3)      Too many figures. Remove unnecessary figures.

4)      Discussion is not sufficient.  

5)      The introduction does not provide a clear understanding of the non-technical reader. Please elaborate more on the aspect of proppant embedment. 

6)   Add more recent references on the behaviour of proppants.

For example:

Bandara, K.M.A.S., Ranjith, P.G. and Rathnaweera, T.D., 2019. Improved understanding of proppant embedment behavior under reservoir conditions: A review study. Powder Technology.

Miedzińska, D., 2019. Numerical modeling of porous ceramics microstructure. Technical Sciences22(1), pp.5-17.

Bandara KM, Ranjith PG, Rathnaweera TD, Perera MS, Kumari WG. Thermally-induced mechanical behaviour of a single proppant under compression: Insights into the long-term integrity of hydraulic fracturing in geothermal reservoirs. Measurement. 2018 May 1;120:76-91.

Ba Geri M, Imqam A, Bogdan A, Shen L. Investigate The Rheological Behavior of High Viscosity Friction Reducer Fracture Fluid and Its Impact on Proppant Static Settling Velocity. InSPE Oklahoma City Oil and Gas Symposium 2019 Apr 8. Society of Petroleum Engineers.

Zheng W, Tannant DD. Influence of proppant fragmentation on fracture conductivity-Insights from three-dimensional discrete element modeling. Journal of Petroleum Science and Engineering. 2019 Mar 9.

7)      Authors need to synthesize results better in their presentation.

8)      Grammatical errors were found throughout the manuscript. Please amend.

Author Response

Comments and Suggestions for Authors

Studies of fracture damage caused by proppant embedment phenomenon in shale rock

The theme of the paper fits very well with this scope of the journal, and it is helpful to broaden the influence of this international journal. This paper provides a clear insight into this field of study by incorporating an experimental approach to understand the impact of proppant embedment on fracture width reduction. However, authors need to carry out the following improvements to the manuscript in order to achieve the publishing stage.

Comments:

1)      Fig. 1 is not clear.

Figure no. 1 has been corrected. (on the page no. 1.)

2) Width reduction can occur as a result of proppant crushing and proppant rearrangement within the fracture. Did you account these factors in your calculations? Please explain as it is not clear.

Yes, we accounted these factors in our calculations. The maximum fracture width (Wfm = 1.514 mm) obtained in the test no. 2, takes into account the width reduction which can occur as a result of proppant crushing and proppant grains rearrangement within the fracture.

Proper sentences with explanation’s ware added (page no. 11 (lines 442 - 448)).

3)      Too many figures. Remove unnecessary figures.

The figure no. 3 on the page no. 2, was delated. All other figures presented in the paper showed graphically the idea and method of performing the research. This will allow the reader to properly understand the research method of the embedment phenomenon.

4)      Discussion is not suffic00ient.  

The conclusions has been expressed on the line no. 501 - 505.

5)      The introduction does not provide a clear understanding of the non-technical reader. Please elaborate more on the aspect of proppant embedment. 

The aspect of proppant embedment is presented on the improved figures no. 2 and 3 as well as on the lines 113 – 129.

6)   Add more recent references on the behaviour of proppants.

For example:

Bandara, K.M.A.S., Ranjith, P.G. and Rathnaweera, T.D., 2019. Improved understanding of proppant embedment behavior under reservoir conditions: A review study. Powder Technology.

Miedzińska, D., 2019. Numerical modeling of porous ceramics microstructure. Technical Sciences22(1), pp.5-17.

Bandara KM, Ranjith PG, Rathnaweera TD, Perera MS, Kumari WG. Thermally-induced mechanical behaviour of a single proppant under compression: Insights into the long-term integrity of hydraulic fracturing in geothermal reservoirs. Measurement. 2018 May 1;120:76-91.

Ba Geri M, Imqam A, Bogdan A, Shen L. Investigate The Rheological Behavior of High Viscosity Friction Reducer Fracture Fluid and Its Impact on Proppant Static Settling Velocity. InSPE Oklahoma City Oil and Gas Symposium 2019 Apr 8. Society of Petroleum Engineers.

Zheng W, Tannant DD. Influence of proppant fragmentation on fracture conductivity-Insights from three-dimensional discrete element modeling. Journal of Petroleum Science and Engineering. 2019 Mar 9.

More literature has been added, as follow:

[27] Bandara, K.M.A.S.; Ranjith, P.G.; Rathnaweera, T.D.. Improved understanding of proppant embedment behavior under reservoir conditions: A review study. Powder Technology 352 (2019), pp. 170-192.

[12] Ghaithan, A.; Al-Muntasheri; Aramco Research Centers-Houston & Saudi Aramco. A Critical Review of Hydraulic Fracturing Fluids over the Last Decade. Paper prepared at the SPE Western North American and Rocky Mountain Joint Regional Meeting held in Denver, Colorado, USA, 16–18 April 2014. SPE 169552-MS. https://doi.org/10.2118/169552-MS

[13] Coronado, J. A. Success of Hybrid Fracs in the Basin. Paper Society of Petroleum Engineers. Production and Operations Symposium, 31 March-3 April 2007, Oklahoma City, Oklahoma, U.S.A. SPE-106758-MS. https://doi.org/10.2118/106758-MS

[14] Handren, P.; Palisch, T. Successful Hybrid Slickwater Fracture Design Evolution – An East Texas Cotton Valley Taylor Case History. Paper  was  presentation  at  SPE  Annual  Technical  Conference  and  Exhibition,  Anaheim,  California,  11–14 November 2007. SPE 110 451. https://doi.org/10.2118/110451-MS

[15] Cawiezel, K.E.; Gupta, D.V.S. Successful Optimization of Viscoelastic Foamed Fracturing Fluids With Ultra lightweight Proppants for Ultralow-Permeability Reservoirs. Journal of SPE Production & Facilities, 25(1): 80–88. 2010. DOI: 10.2118/119626-PA.

7)      Authors need to synthesize results better in their presentation.

Some figures has been changed and improved, ex: the red profile line to the white line, scale in Fig. no. 11, 12, 13 and 14, etc, to be more untestable by readers.

8)      Grammatical errors were found throughout the manuscript. Please amend.

The grammatical errors in the manuscript has been corrected by native speaker.

Reviewer 2 Report

The work by Masłowski et al. is an interesting experimental study that they investigated the degree of formation damage due to proppant embedment. The results are valuable that concerns of this work are found below.

1.       Kindly to introduce more background of hydraulic fracturing in unconventional reservoirs. The following articles are recommended to be consulted. SPE-169552-MS; Fuel 2019, 236:404-427.

2.       Please state clearly the initiatives of this research in the introduction. List objectives clearly in number at the end of the introduction.

3.       Fig. 11-14 have a black background which is not suitable for b&w print; it is better to change to the white background if there is a way. Scale in Fig. 14 is not clear to see; please check.

4.       Line 120, please do not use “0.600÷0.300”, simply use the result or include the result in the formula.

5.       It is recommended to add scale as well in real pictures like Fig. 5a.

6.       It is not very clear how the authors measure the fracture width practically; please stress this point by giving more details in the revised version.

7.       Fig. 8’s scale needs more resolution.

8.       What proppant the authors use? Please clarify by giving more details.

9.       The author discussed fracture width in different scenarios, but no fracture/matrix porosity and permeability are reported; they are important petrophysical properties for fracture. It is recommended to discuss the extended study of this one based on the following articles. JNGSE 2018, 58:216-233; J Unconv Oil Gas 2015, 11: 27-43; SPE J 2018, 23(04): 1452-1468. SPE-17655-PA; SPE-60184-JPT; TIPM 1996, 23(01): 1-30.

10.   It is suggested to express the conclusions in bullets.

Author Response

Comments and Suggestions for Authors

The work by Masłowski et al. is an interesting experimental study that they investigated the degree of formation damage due to proppant embedment. The results are valuable that concerns of this work are found below.

1.       Kindly to introduce more background of hydraulic fracturing in unconventional reservoirs. The following articles are recommended to be consulted. SPE-169552-MS; Fuel 2019, 236:404-427.

More background knowledge of hydraulic fracturing in unconventional reservoirs has been added (on the page no. 2, lines 44 – 56).

for example:

[12] Ghaithan, A.; Al-Muntasheri; Aramco Research Centers-Houston & Saudi Aramco. A Critical Review of Hydraulic Fracturing Fluids over the Last Decade. Paper prepared at the SPE Western North American and Rocky Mountain Joint Regional Meeting held in Denver, Colorado, USA, 16–18 April 2014. SPE 169552-MS. https://doi.org/10.2118/169552-MS

[13] Coronado, J. A. Success of Hybrid Fracs in the Basin. Paper Society of Petroleum Engineers. Production and Operations Symposium, 31 March-3 April 2007, Oklahoma City, Oklahoma, U.S.A. SPE-106758-MS. https://doi.org/10.2118/106758-MS

[14] Handren, P.; Palisch, T. Successful Hybrid Slickwater Fracture Design Evolution – An East Texas Cotton Valley Taylor Case History. Paper  was  presentation  at  SPE  Annual  Technical  Conference  and  Exhibition,  Anaheim,  California,  11–14 November 2007. SPE 110 451. https://doi.org/10.2118/110451-MS

[15] Cawiezel, K.E.; Gupta, D.V.S. Successful Optimization of Viscoelastic Foamed Fracturing Fluids With Ultra lightweight Proppants for Ultralow-Permeability Reservoirs. Journal of SPE Production & Facilities, 25(1): 80–88. 2010. DOI: 10.2118/119626-PA.

[27] Bandara, K.M.A.S.; Ranjith, P.G.; Rathnaweera, T.D.. Improved understanding of proppant embedment behavior under reservoir conditions: A review study. Powder Technology 352 (2019), pp. 170-192.

2.       Please state clearly the initiatives of this research in the introduction. List objectives clearly in number at the end of the introduction.

The initiatives of this research in the introduction has been added, on the page no. 3 (lines 130 - 136).

3.       Fig. 11-14 have a black background which is not suitable for b&w print; it is better to change to the white background if there is a way. Scale in Fig. 14 is not clear to see; please check.

Change the black background to the white background is unfortunately impossible. The red profile line to the white line was improved as well as the scale in Fig. no. 11, 12, 13 and 14.

4.       Line 120, please do not use “0.600÷0.300”, simply use the result or include the result in the formula.

Hopefully this will be more clear: “from 0.600 to 0.300 mm” on the page no. 3 (line 141).

5.       It is recommended to add scale as well in real pictures like Fig. 5a.

The scale in real pictures Fig. 4a and 4b (old Figure 5a and 5b.), on the page no. 4. was added.

6.       It is not very clear how the authors measure the fracture width practically; please stress this point by giving more details in the revised version.

The fracture width Wf is measure within equations (7), also lines no. 249 – 253, 264 – 271 explained the methodology of measurements and results are presented on the figure 14 and 15.

7.       Fig. 8’s scale needs more resolution.

We do not have Fig. 7 (old Fig. 8) with a higher resolution.

8.       What proppant the authors use? Please clarify by giving more details.

More details about proppant are on the page no. 4 (lines 142 - 143).

9.       The author discussed fracture width in different scenarios, but no fracture/matrix porosity and permeability are reported; they are important petrophysical properties for fracture. It is recommended to discuss the extended study of this one based on the following articles. JNGSE 2018, 58:216-233; J Unconv Oil Gas 2015, 11: 27-43; SPE J 2018, 23(04): 1452-1468. SPE-17655-PA; SPE-60184-JPT; TIPM 1996, 23(01): 1-30.

In the article, the authors did not investigate the porosity of the matrix and its permeability as well as petrophysical properties as important fracture properties. They focused on determining the quantities characteristic of the embedment phenomenon, such as the depth and width of embed grains into the fracture wall and their effect on the decrease in the gap width Wf.

10.   It is suggested to express the conclusions in bullets.

I expressed the conclusions on the lines no. 501 - 505.

Round 2

Reviewer 1 Report

The authors have carefully evaluated the recommendation for revision and made the necessary changes according to the suggestions. The revised version is suitable for the publication in Applied sciences. 

Reviewer 2 Report

The authors addressed most of the issues, therefore it is recommended to be published.